# Pre-training Text-to-Text Transformers for Concept-centric Common Sense

**Wangchunshu Zhou**[1]*, **Dong-Ho Lee**[2]*, **Ravi Kiran Selvam**[2], **Seyeon Lee**[2],
**Bill Yuchen Lin**[2], **Xiang Ren**[2]
[1] Beihang University [2] University of Southern California
zhouwangchunshu@buaa.edu.cn, {dongho.lee, xiangren}@usc.edu

## Abstract

Pre-trained language models (PTLM) have achieved impressive results in a range of natural language understanding (NLU) and generation (NLG) tasks. However, current pre-training objectives such as masked token prediction (for BERT-style PTLMs) and masked span infilling (for T5-style PTLMs) do not explicitly model the relational commonsense knowledge about everyday concepts, which is crucial to many downstream tasks that need common sense to understand or generate. To augment PTLMs with concept-centric commonsense knowledge, in this paper, we propose both *generative* and *contrastive* objectives for learning common sense from the text, and use them as *intermediate* self-supervised learning tasks for incrementally pre-training PTLMs (before task-specific fine-tuning on downstream datasets). Furthermore, we develop a joint pre-training framework to unify generative and contrastive objectives so that they can mutually reinforce each other. Extensive experimental results show that our method, *concept-aware language model* (**CALM**)[1], can pack more commonsense knowledge into the parameters of a pre-trained text-to-text transformer *without* relying on external knowledge graphs, yielding better performance on both NLU and NLG tasks. We show that while only incrementally pre-trained on a relatively small corpus for a few steps, CALM outperforms baseline methods by a consistent margin and even comparable with some larger PTLMs, which suggests that CALM can serve as a general, "plug-and-play" method for improving the commonsense reasoning ability of a PTLM.

## 1 Introduction

Pre-trained language models (PLTMs) such as BERT (Devlin et al., 2018) and T5 (Raffel et al., 2019) have revolutionized the field of NLP, yielding impressive performance on various conventional natural language understanding (NLU) and generation (NLG) tasks. BERT and its novel variants such as RoBERTa (Liu et al., 2019) and ALBERT (Lan et al., 2019) capture syntactical and semantic knowledge mainly from the pre-training task of *masked language modeling*, while T5-style models such as BART (Lewis et al., 2019) instead focus on *masked span infilling* tasks. Though yielding better performance on many downstream tasks, these pre-training objectives, however, do not explicitly guide the models to reason with *concept-centric* commonsense knowledge from language, including the relation and composition of daily concepts in our lives. This leaves room for equipping current PTLMs with richer commonsense reasoning ability.

For example, consider a multi-choice question "*What do you fill with ink to write notes on a piece of copy paper? (A) fountain pen (B) pencil case (C) printer (D) notepad*". The current state-of-the-art question answering model, UnifiedQA (Khashabi et al., 2020), which was fine-tuned on T5-large with multiple datasets, still predicts '(C) *printer*' as its answer. The model may be overly sensitive to the *co-occurrence* between phrases in question sentence like '*ink*' and '*copy paper*' and the answer choice '*printer*', but fails to reason with the concept-centric knowledge that '*fountain pen*' is a writing instrument that needs to be filled with '*ink*'. Such mistake in commonsense reasoning becomes a bottleneck for current PTLMs (Davis & Marcus, 2015). Towards augmenting PTLMs with more

---

* Equal contribution. The work was done when Wangchunshu was visiting USC.
[1]Code will be published at: https://github.com/INK-USC/CALM

knowledge, prior works mainly focus on training larger models (Brown et al., 2020), adding specific architectures to exploit external knowledge (Peters et al., 2019), or incorporating knowledge bases for pre-training (Xiong et al., 2020). In this paper, we instead look to explicitly *teach* pre-trained models to write and reason with common concepts through novel pre-training strategies.

We present two kinds of self-supervised pre-training tasks: **concept-to-sentence generation (C2S)** and **concept order recovering (COR)**. C2S trains the pre-trained model to compose ("write") sentences given a set of concepts, and expects the generated sentences to be fluent and plausible in terms of commonsense. COR aims to teach models to detect and revise a corrupted sentence with incorrect ordering of concepts. As illustrated in Figure 1, both tasks require a pre-trained model to recall relevant commonsense facts about the concepts and to understand the underlying commonsense relations between them. Both of the proposed objectives can explicitly encourage the model to capture the relational concept-centric commonsense knowledge and perform compositional reasoning.

Specifically, we need a generative pre-training objective to encourage models to capture this **generative commonsense** reasoning ability, so that models can learn to generate sentences with commonsense knowledge for both C2S and COR. Also, to teach modes to distinguish truth sentences from less plausible ones, we need to teach models with **discriminative commonsense** through *contrastive self-training*. To unify both *generative* and *contrastive* objectives within a joint learning framework so that the model can learn both generative and discriminative commonsense knowledge at the same time, we propose to use the sentences generated by the model itself as the distractors and train the model to distinguish the generated sentences from real sentences. In this way, the model is forced to acquire new commonsense knowledge in order to distinguish the distractors generated by itself, which probably exploit the knowledge the model already possesses. Therefore, the model is trained to iteratively improve upon itself in a self-play fashion. We share all the parameters between the generator (trained with the generative objective) and the discriminator (trained with the contrastive objective), then train multiple objectives with different prefixes. Compared to previous works (Peters et al., 2019; Li et al., 2019; Xiong et al., 2020) that utilize external knowledge bases like Wikidata or ConceptNet, our approach can directly improve the generative and discriminative commonsense reasoning ability of PTLMs at the same time without relying on external knowledge bases.

To evaluate the effectiveness of our proposed method, we apply our method in an intermediate-task transfer learning setting (Pruksachatkun et al., 2020) based on the pre-trained T5-base model to train a *Concept-Aware Language Model* (**CALM**). While only continually pre-trained on a small dataset for a relatively fewer number of updates (compared to conventional pre-training), CALM consistently outperforms T5-base on four commonsense-related NLU datasets (i.e., COMMONSENSEQA, OPEN-BOOKQA, PIQA, and ANLI) and COMMONGEN, a commonsense-related NLG dataset. Our results and careful ablation studies demonstrate the potential of our method to serve as a "plug-and-play" method for any pre-trained text-to-text transformer before fine-tuning on commonsense-related tasks. To the best of our knowledge, our work is the first to investigate concept-centric self-supervised objectives that improve both generative and discriminative commonsense reasoning ability of a pre-trained language model.

## 2 SELF-SUPERVISED OBJECTIVES FOR CONCEPT-CENTRIC LEARNING

In this section, we first describe the proposed generative and contrastive objectives used for improving the commonsense reasoning ability of pre-trained text-to-text transformers. Then, we introduce the joint learning framework which unifies the proposed self-supervised objectives and learn a unified text-to-text transformer based on pre-trained models such as T5.

### 2.1 GENERATIVE OBJECTIVES

Similar to many other pre-training tasks such as masked language modeling, we aim to teach models to recover original sentences from corrupted inputs, which is often regarded as a *denoising* process. We propose two generative self-supervised pre-training objectives: concept-to-sentence generation (C2S) and concept order recovering (COR).

**Concept Extraction.** Given an input $\mathbf{x} = [x_1, x_2, \ldots, x_n]$, we first conduct part-of-speech tagging with Spacy for the sentence and extract *Verb*, *Noun*, and *Proper Nouns* from the sentence to use as

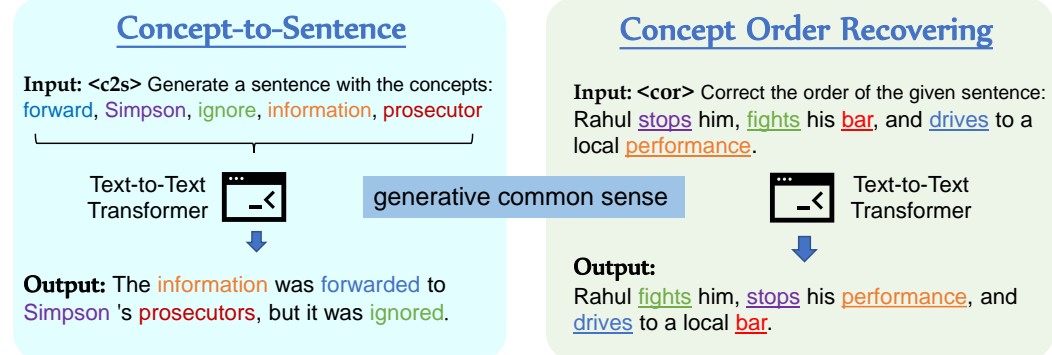

Figure 1: **Two self-supervised pre-training objectives that teach text-to-text transformers with generative common sense:** (1) **Concept-to-Sentence Generation** (C2S) pre-trains the model to recover the original sentence with a shuffled concept set, e.g., {*forward*, *Simpson*, *ignore*, *information*, *prosecutor*} → "The information was forwarded to Simpson's prosecutors, but it was ignored." (2) **Concept Order Recovering** (COR), similarly, teaches the model to correct the mispositioned concepts in the original sentence. For example, the concepts (*stops*, *fights*, *bar*, *drives*, *performance*), are randomly reordered in the input, while the model should recover the original sentence.

concepts[2]. Next, we form concept-sets $\mathcal{C} = [v_1, v_2, \ldots, v_p, n_1, n_2 \ldots, n_q]$ where $v_i$ and $n_i$ denotes the i-th verb or noun/proper noun concept (token) in $\mathbf{x}$. We denote $\mathcal{C}_v$ and $\mathcal{C}_n$ as the set of verb and noun/proper noun concepts respectively in $\mathcal{C}$. (i.e. $\mathcal{C}_v = [v_1, v_2, \ldots, v_p]$ and $\mathcal{C}_n = [n_1, n_2, \ldots, n_q]$.)

**Concept-to-Sentence Generation (C2S).** The concept-to-sentence generation (C2S) objective requires the text-to-text transformer to recover the original sentence given only a few unordered keywords of the sentence. Specifically, given a sentence, we shuffle the extracted concept-set $\mathcal{C}$ to create the perturbed source sequence and train the model to generate the original sentence with a prefix (denoted as `<c2s>`) as described in Fig. 1. Formally, the C2S objective can be formulated as:

$$L_{c2s} = \mathbb{E}\Big( \sum_{i=1}^{n} -\log p(x_i | \texttt{<c2s>}; \text{PERMUTE}(\mathcal{C}); x_{1:i-1}) \Big) \tag{1}$$

where the PERMUTE() function randomly shuffle the concepts in the concept-set. This objective requires the model to construct an acceptable commonsense sentence by adhering to and reasoning over the commonsense relations between the given concepts. Therefore, relational commonsense knowledge is implicitly injected into the parameters of the model. The C2S objective is motivated by the task proposed in Lin et al. (2020). Compared to their work, the concept-set used in C2S covers more concepts such as named entities, while the original task only includes the concepts appearing in ConceptNet. We apply the task in a general domain and as a pre-training objective, instead of merely serving as an evaluation task.

**Concept Order Recovering (COR).** As for the concept order recovering (COR) objective, we shuffle the order of concept in a sentence and train the model to recover the original sentence. As illustrated in Figure 1, given an input sentence "tree grows on the apple,", the models would shuffle the concepts including "tree", "grow", and "apple" to recover the original sentence "apple grows on the tree." The noise introduced by concept shuffling is different from that by traditional self-supervised objectives like mask language modeling and mask span prediction because the corrupted source sentences are in general complete (i.e., no tokens or spans are masked) and grammatically correct, while not acceptable in terms of commonsense because the order and relation between concepts are shuffled. By training the model to detect and correct the disorder of concepts in a sentence, the model is expected to acquire some relational commonsense knowledge like "apple generally grows on a tree" instead of "tree grows on an apple."

Formally, the COR objective can be formulated as:

$$L_{cor} = \mathbb{E}\Big( \sum_{i=1}^{n} -\log p(x_i | \texttt{<cor>}; \text{CONCEPT-PERMUTE}(\mathbf{x}, \mathcal{C}); x_{1:i-1}) \Big), \tag{2}$$

---

[2]We split the concepts with multiple tokens (under Spacy tokenization) into single token to ensure the concepts discussed afterwards all contain a single token.

where `<cor>` is the prefix for the COR objective illustrated in Figure 1. The function CONCEPT-PERMUTE() permutes the order between concepts in the same category (i.e. noun or verb) in the sentence, which can be formally defined as:

$$\text{CONCEPT-PERMUTE}(\mathbf{x}, \mathcal{C}) = [x_1', x_2', \ldots, x_n'] \text{ where } x_i' = \begin{cases} x_i & x_i \notin \mathcal{C} \\ \text{PERMUTE}(\mathcal{C}_v)[j] & x_i = v_j \\ \text{PERMUTE}(\mathcal{C}_n)[j] & x_i = n_j \end{cases} \quad (3)$$

Our proposed objectives require the model to capture the relational commonsense knowledge between concepts and perform relational (COR) and compositional (C2S) commonsense reasoning in order to successfully reconstruct the original sentence. Therefore, the model is encouraged to acquire concept-centric commonsense knowledge more effectively. In contrast, conventional pre-training objectives like masked language modeling and masked span infilling mainly focus on general token-level co-occurrence patterns and thus are less effective for learning commonsense knowledge.

## 2.2 CONTRASTIVE OBJECTIVE

The contrastive objective encourages the pre-trained model to distinguish the real sentence from a distractor sentence: a sentence that is similar to the real sentence, generally grammatically correct, but may not follow common sense. We expect it to improve the pre-trained model's discriminative commonsense reasoning ability so that the model's performance on commonsense-reasoning-discriminative tasks, like CommonsenseQA, can be improved. We formulate the contrastive objective as a **Generative QA** task: we take the concatenation of a prefix `<cont>` (question / context), the real sentence $x$ (answer), and the distractor $x'$ (distractor) as the input and train the model to output the real sentence $x$. Formally, we have the loss function of the contrastive objective defined as:

Figure 2: **Overview of Contrastive self-supervised pre-training objectives.** Generative QA style contrastive objective requires the model to distinguish truth sentences from less plausible ones.

$$L_{cont} = \mathbb{E}\big( - \log p(x | \texttt{<cont>}; \text{PERMUTE}(x; x'))\big), \quad (4)$$

where the prefix `<cont>` is described in Figure 2. The distractor $x'$ is either constructed by concept shuffling as described previously (i.e. $x' = \text{CONCEPT-PERMUTE}(\mathbf{x}, \mathcal{C})$) when used independently, or generated by a generator trained with the aforementioned generative objectives when used in the joint training framework, which will be described in the next section.

## 3 JOINT TRAINING WITH GENERATIVE AND CONTRASTIVE OBJECTIVES

The aforementioned generative and contrastive self-supervised objectives can be applied independently or simply combined in a multi-task learning fashion. We argue that these two objectives can mutually reinforce each other: the generated sentences from the generative objective can help the contrastive module learn to distinguish commonsense sentences from less plausible ones.

Therefore, we propose a joint training framework to unify generative objectives and contrastive objectives by using the generator to produce *distractors* for learning towards contrastive objective.

Specifically, we have a **generator** $G_\theta$ (trained with the generative objectives) and a **discriminator** $D_\phi$

**Algorithm 1:** Pre-training Concept-Aware Language Model (CALM).

**Input:** Text-to-Text Transformer $T_\theta$, Text corpus X=[$x_1, x_2, \ldots, x_n$].

**repeat**
  **for** *each $x_i \in X$* **do**
    Extract the concept-set $\mathcal{C}_i$;
    Construct the distractor sentence
      $x' = \text{CONCEPT-PERMUTE}(\mathbf{x}_i, \mathcal{C}_i)$;
    Update $T_\theta$ with Eq.(1, 2, 4);
**until** *maximum iterations reached*;
**repeat**
  **for** *each $x_i \in X$* **do**
    Update $T_\theta$ with Eq.(7)
**until** *maximum iterations reached*;

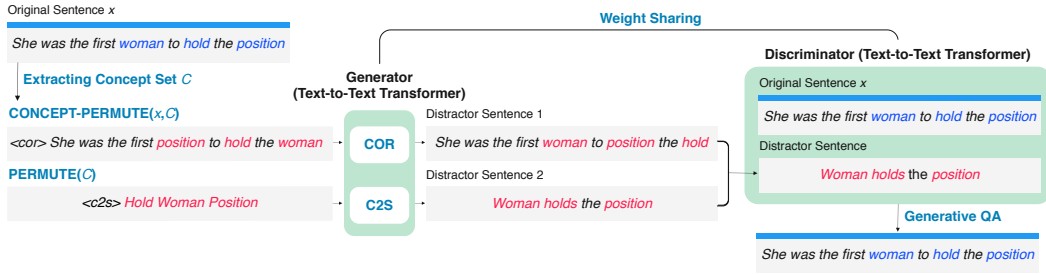

Figure 3: **Proposed Joint Training Framework.** Given an input sentence $x$ ("She was the first *woman* to *hold* the *position*."), we extract concept-set $\mathcal{C}$ (*woman*, *hold*, *position*). Given $x$ and $\mathcal{C}$, we produce corrupted source sequence $x'$ either for C2S and COR. The generator trained with the corresponding objective recovers sentences as distractors $x''$ to the discriminator. The discriminator is trained to distinguish truth sentences from randomly selected distractor among two objectives. Parameters between the generator and discriminator are shared.

(trained with the contrastive objective). Given an input sentence $\mathbf{x}$, we first use the method for either C2S or COR to produce the corrupted source sequence $\mathbf{x}'$. Then, we use the generator $G_\theta$ trained with the corresponding objective to generate the recovered sentence $\mathbf{x}'' = G_\theta(x')$. We then take $\mathbf{x}''$ as the distractor to train the discriminator $D_\phi$ with the contrastive objective. The loss function of the proposed joint training framework consists of two parts: the first part is the loss of generative objectives, which is identical to the loss described in Eq.(1) and Eq.(2) and is used to update the generator $G_\theta$. The second part is the loss of the contrastive objective as described in Eq.(4), which can be formulated as:

$$L_{cont\_joint\_c2s} = \mathbb{E}\big(-\log \mathbf{D}_\phi(y|\texttt{<cont>}; x; \mathbf{G}_\theta(\texttt{<c2s>}; \text{PERMUTE}(\mathcal{C})))\big) \quad (5)$$

$$L_{cont\_joint\_cor} = \mathbb{E}\big(-\log \mathbf{D}_\phi(y|\texttt{<cont>}; x; \mathbf{G}_\theta(\texttt{<cor>}; \text{CONCEPT-PERMUTE}(\mathbf{x}, \mathcal{C})))\big) \quad (6)$$

where $L_{cont\_joint\_c2s}$ and $L_{cont\_joint\_cor}$ is the contrastive loss with the distractor generated with either the C2S or the COR objective and $y$ is the original sentence. We then have the overall objective for the joint training framework defined as :

$$L_{joint} = (L_{c2s} + L_{cor}) + \beta(L_{cont\_joint\_c2s} + L_{cont\_joint\_cor}). \quad (7)$$

$L_{c2s}$ and $L_{cor}$ are defined in Eq.(1) and Eq.(2) respectively and $\beta$ is a hyperparameter controlling the relative weight between the generative and contrastive objectives. Note that since we would like to inject both generative and discriminative commonsense reasoning ability into the parameters of a single text-to-text transformer, we share the parameters between the generator $G_\theta$ and the discriminator $D_\phi$.

Finally, we describe the overall procedure to apply the proposed self-supervised objectives and the joint training framework on a pre-trained text-to-text transformer. We apply a two-stage training strategy. During the first stage, we apply our proposed generative and contrastive objectives individually on the model in a multi-task learning fashion with different prefixes. This provides a good starting point for the second stage where the joint training framework is applied. We summarize the workflow of our method in Algorithm 1.

## 4 EXPERIMENTS

In this section, motivated by the observation of Pruksachatkun et al. (2020) that tasks requiring commonsense reasoning ability generally serve as good intermediate task, we test our method in the intermediate task transfer setting. Specifically, we initialize our model with T5-base, a pre-trained text-to-text transformer model, and training the model with our proposed method as intermediate task before fine-tuning and target downstream tasks. Another reason for adopting this setting is because we expect our method to serve as a "plug-and-play" method that can be applied to any pre-trained text-to-text transformer by simply continually training for a few steps.

**Details for Pre-training and Fine-tuning** CALM is continually pre-trained with our proposed self-supervised objectives as intermediate tasks based on the pre-trained T5-base model following

the setting in Pruksachatkun et al. (2020). We randomly sample 500K sentences from the English Wikipedia corpus[3], which is used for pre-training BERT and its variants, as the source dataset for our proposed self-supervised objectives which serve as intermediate tasks. We then fine-tune the CALM on each downstream task individually and report the average performance of three runs with different random seeds for fine-tuning on each dataset since the performance is sensitive to different random seeds. Training details and hyperparameter settings are presented in Appendix A.1 and A.2.

**Datasets** We consider five commonsense benchmark datasets as target tasks. We categorize these datasets into discriminative and generative tasks. Discriminative tasks are classification tasks while generative tasks are text generation tasks. We consider four datasets for discriminative task: **CommonsenseQA** (Talmor et al., 2018), **OpenbookQA** (Mihaylov et al., 2018), **PIQA** (Bisk et al., 2020), **aNLI** (Bhagavatula et al., 2019) and one dataset for generative task: **CommonGEN** (Lin et al., 2020). Details on datasets are discussed in Appendix A.3.

**Compared Methods** We compare our model with following models continually trained with different intermediate tasks based on the pre-trained T5-base model: (1) **T5-base** is the pre-trained T5-base model without continually training on any intermediate task. (2) **T5-base w/ additional epochs** is continually pre-trained using the original pre-training objective of T5 with additional training steps. The total number of additional training steps is equal to that of our final model. (3) **T5-base + SSM** is continual pre-trained with a variant of the *salient span masking* objective (Guu et al., 2020; Roberts et al., 2020) objective that masks text spans of concepts extracted with POS tagging instead of named entities extracted by a pre-trained NER model, which makes it more focused on concepts. (4) **CALM(Generative-Only)** is continually pre-trained with the proposed generative objectives including concept-to-sentence generation(C2S) and concept order recovering(COR) as intermediate tasks. (5) **CALM(Contrastive-Only)** is continually pre-trained with the proposed contrastive objective as described in section 2.2 using the distractor generated by concept shuffling. (6) **CALM(Mix-only)** is continually pre-trained with both the generative objectives and the contrastive objective, combined with a multi-task learning fashion with identical weights for each objective as the intermediate task. (7) **CALM (w/o Mix warmup)** is continually pre-trained with the joint training objective described in Eq (7) directly from the pre-trained T5-base model. (8) **CALM** is our main model trained as described in Algorithm 1. The difference between CALM and CALM (Joint) is that the former is initialized by the CALM(Mix). We also include the performance of the BERT-base model and two knowledge enhanced PTLMs that have similar architecture to BERT-base.

**Evaluation Metrics** For discriminative tasks, we choose accuracy as our metric following other conventional question answering tasks. For generative tasks, we report automated metrics including BLEU (Papineni et al., 2002), METEOR (Banerjee & Lavie, 2005), CIDEr (Vedantam et al., 2015), and SPICE (Anderson et al., 2016) following the leaderboard of COMMONGEN (Lin et al., 2020). Results for COMMONGEN are on the test set and others are on the official development set. We tune the hyperparameters based on the models' performance on a in-house split dev set.

## 4.1 EXPERIMENTAL RESULTS

The result is presented in Table 1. First, we can see that our CALM model consistently and significantly (with p-value $< 0.01$) outperforms the backbone T5-base model on all five datasets by a margin range from 1.5 to 2.9 accuracy on discriminative tasks and 1.5/0.6 BLEU/SPICE score on CommonGEN. This is an impressive result since we are only performing intermediate training on a relatively small dataset for only around 20k updates. It demonstrates the potential of our method for serving as a "plug-and-play" method for packing more commonsense knowledge into a pre-trained text-to-text transformer. Table 3 also shows that CALM performs comparably with several large-size PTLMs like BART, T5-large, and GPT-2 on the COMMONGEN dataset. The performance is worse than KG-BART (Liu et al., 2020), the current state-of-the-art on COMMONGEN, which is a contemporary work that exploits external knowledge bases as additional information, and is based on a larger backbone(i.e., BART (Lewis et al., 2019)).

In addition, we can observe that both the proposed generative and contrastive objective outperforms the backbone T5-base model, as well as its variants that continually pre-trained with the original masked span prediction objective and the concept-specific salient span masking scheme, when applied independently. Note that we find the variant of salient span masking that focuses on concept is not

---

[3]https://dumps.wikimedia.org/enwiki/latest/

| Methods | CSQA | OBQA | PIQA | aNLI | CommonGEN | | | |
|---|---|---|---|---|---|---|---|---|
| | Accuracy (official dev) | | | | BLEU-4 | METEOR | CIDEr | SPICE |
| BERT-base | 53.08(±0.16) | 57.60(±0.8) | 64.86(±0.52) | 61.88(±0.56) | - | - | - | - |
| ERNIE | 54.06(±0.12) | 58.90(±0.9) | 66.47(±0.58) | 63.04(±0.46) | - | - | - | - |
| KnowBERT | 53.88(±0.15) | 58.50(±0.8) | 66.61(±0.63) | 63.18(±0.52) | - | - | - | - |
| T5-base | 61.88(±0.08) | 58.20(±1.0) | 68.14(±0.73) | 61.10(±0.38) | 24.90 | 31.20 | 12.99 | 32.40 |
| T5-base + cont. pretraining | 61.92(±0.45) | 58.10(±0.9) | 68.19(±0.77) | 61.15(±0.52) | 25.10 | 31.00 | 13.12 | 32.40 |
| T5-base + SSM | 62.08(±0.41) | 58.30(±0.8) | 68.27(±0.71) | 61.25(±0.51) | 25.20 | 31.20 | 13.28 | 32.40 |
| CALM (Generative-Only) | 62.28(±0.36) | 58.90(±0.4) | 68.91(±0.88) | 60.95(±0.46) | 25.80 | 31.20 | 13.81 | 32.60 |
| CALM (Contrastive-Only) | 62.73(±0.41) | 59.30(±0.3) | 70.67(±0.98) | 61.35(±0.06) | 25.50 | 31.20 | 13.58 | 32.60 |
| CALM (w/o Mix warmup) | 62.18(±0.48) | 59.00(±0.5) | 69.21(±0.57) | 61.25(±0.55) | 25.80 | 31.20 | 13.77 | 32.60 |
| CALM (Mix-only) | 63.02(±0.47) | 60.40(±0.4) | 70.07(±0.98) | 62.79(±0.55) | 26.00 | 31.20 | 13.82 | 32.80 |
| CALM | **63.32(±0.35)** | **60.90(±0.4)** | **71.01(±0.61)** | **63.20(±0.52)** | **26.40** | **31.40** | **13.88** | **33.00** |

Table 1: **Experimental results on commonsense reasoning datasets.** The first group of models are baselines. The models in the middle group and last group except the CALM model are trained with the proposed objectives independently and the final CALM model is trained by joint training. Best models are bold and second best ones are underlined within each metric.

| Methods | CSQA | OBQA | PIQA | aNLI | CommonGEN | | | |
|---|---|---|---|---|---|---|---|---|
| | Accuracy (official dev) | | | | BLEU-4 | METEOR | CIDEr | SPICE |
| BERT-large | 57.06(±0.12) | 60.40(±0.6) | 67.08(±0.61) | 66.75(±0.61) | - | - | - | - |
| T5-large | 69.81(±1.02) | 61.40(±1.0) | 72.19(±1.09) | 75.54(±1.22) | 28.60 | 30.10 | 14.96 | 31.60 |
| CALM-large (Mix-only) | 70.26(±0.23) | 62.50(±1.0) | 73.70(±1.09) | 75.99(±1.26) | 29.20 | 31.30 | 15.24 | 33.10 |
| CALM-large | 71.31(±0.04) | **66.00(±1.0)** | 75.11(±1.65) | 77.12(±0.34) | **29.50** | **31.90** | **15.61** | **33.20** |
| RoBERTa-large[4] | **71.81(±0.25)** | 63.90(±0.8) | **76.90(±0.62)** | **82.35(±0.54)** | - | - | - | - |

Table 2: **Experimental results on large model.** Comparison between large models of other PTLMs and CALM. Best models are bold and second best ones are underlined within each metric.

very effective. We suspect this is because the resulting training data would be somewhat similar to the original text infilling objective because concepts are very common in the corpus and we only train for a few steps. The combination of the generative and contrastive objectives (i.e., CALM(Mix-only)) yields further improvement upon the model trained independently with either generative or contrastive objectives. Also, we find that the CALM model consistently outperforms CALM(Mix), demonstrating the effectiveness of the proposed joint training framework. Applying joint training directly on top of a pre-trained model (i.e., CALM(w/o Mix warmup)) does not work very well, demonstrating the necessity of applying mixed training to initialize the model before starting joint training.

To further confirm the effectiveness of our approach, we also apply our method to continually pre-train T5-large with the same data and number of training steps. We then compare the performance of the resulting model with that of the original T5-large model in Table 10. We find that both the proposed training objectives and the joint training framework consistently and significantly (with p-value $< 0.01$) improve upon the original T5-large, showing our approach is effective for models with different sizes. Our model also outperforms BERT-large by a large margin.

| Methods | Params | CommonGEN | | | |
|---|---|---|---|---|---|
| | | BLEU-4 | METEOR | CIDEr | SPICE |
| GPT-2 (Radford et al., 2019) | 774M | 21.10 | 26.20 | 12.15 | 25.90 |
| UniLM (Dong et al., 2019) | 340M | 27.70 | 29.70 | 14.85 | 30.20 |
| BART (Lewis et al., 2020) | 406M | 26.30 | 30.90 | 13.92 | 30.60 |
| T5-base (Raffel et al., 2019) | 220M | 16.40 | 23.00 | 9.16 | 22.00 |
| T5-large[5] (Raffel et al., 2019) | 770M | 28.60 | 30.10 | 14.96 | 31.60 |
| KG-BART[6] (Liu et al., 2020) | 406M | **30.90** | **32.40** | **16.83** | 32.70 |
| T5-base (our implementation) | 220M | 24.90 | 31.20 | 12.99 | 32.40 |
| CALM-base | 220M | 26.40 | 31.40 | 13.88 | 33.00 |
| CALM-large | 774M | 29.50 | 31.90 | 15.61 | **33.20** |

Table 3: Comparison between PTLMs on CommonGEN. Above baselines are reported number in the leaderboard. T5-base(our implementation) uses different hyperparmeter setting than that reported in the leaderboard.

However, our model performs slightly worse compared to RoBERTa-large. We suspect this is because RoBERTa-large is optimized for more steps than T5-large and our CALM-large. This is also observed in many other tasks and datasets.

### 4.2 Performance Analysis

**Analysis on Generative objective** To investigate the contribution of each generative objective, we conduct an ablation study by continually pre-training three models from the same T5-base model with C2S, COR, and text infilling, which is the original objective for pre-training T5, as the objective for the intermediate task. We continually pre-train these models for the same number of steps and then evaluate their performance by fine-tuning on different target tasks. The result is shown in Table 4.

| Methods | CSQA | PIQA | CommonGEN | | | | Methods | CSQA | PIQA | CommonGEN | | | |
|---|---|---|---|---|---|---|---|---|---|---|---|---|---|
| | Accuracy | | BLEU-4 | METEOR | CIDEr | SPICE | | Accuracy | | BLEU-4 | METEOR | CIDEr | SPICE |
| T5 - Text Infilling | 61.92 | 68.19 | 25.10 | 31.00 | 13.13 | 32.40 | Multi-choice QA | 62.21 | 68.82 | 25.00 | 31.20 | 13.28 | 32.60 |
| CALM - COR | **62.36** | **68.77** | 25.70 | 31.20 | 13.65 | 32.60 | True/False | 62.24 | 67.81 | 25.10 | 31.20 | 13.41 | 32.60 |
| CALM - C2S | 62.24 | 68.75 | **25.90** | **31.40** | **13.94** | **32.80** | Generative QA | **62.73** | **70.67** | **25.50** | 31.20 | **13.58** | 32.60 |

|  (a) Generative objectives | (b) Contrastive objectives |
|---|---|

Table 4: **Analysis on Contrastive and Generative objectives.** Left table shows the performance on downstream tasks by pre-training with different generative objective (COR, C2S, and original objective for pre-training T5). Right table shows the performance on downstream tasks by pre-training with different task formats of contrastive objective.

We can see that both C2S and COR works better than the original masked span infilling objective on itself. This confirms the effectiveness of our proposed generative objectives on improving the commonsense reasoning ability of pre-trained text-to-text transformers.

**Task Formulation of the Contrastive objectives** For contrastive objectives, we test three different task formats: Multi-choice QA, Generative QA, and True/False. Multi-choice QA and Generative QA takes the concatenation of the real sentence and the distractor. Then, Multi-choice QA output the index of the real sentence following other conventional Multi-choice QA tasks, and Generative QA output the real sentence respectively. True/False takes either the real sentence or the distractor and train the model to perform a binary classification problem of whether the input sentence makes sense. The result is shown in Table 4. We could find that the format of Generative QA performs the best. We suspect this is because the Generative QA format is closer to the format used during the original pre-training stage of the T5 model and the format used for fine-tuning.

**Performance with fewer training examples** To investigate the effectiveness of our objective in the low-resource setting, we explore the performance of our model and baselines fine-tuning with different fractions of the training data. From Figure 4, we can see that the performance improvement yielded by our models upon the T5-base model is more significant in the low-resource regime. This shows that CALM may already pack some

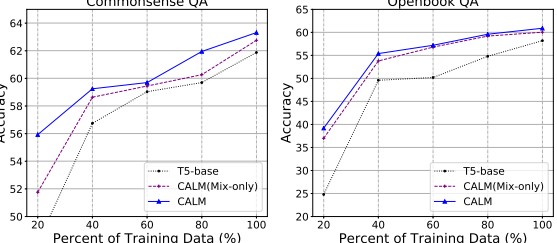

Figure 4: Performance of compared models fine-tuned with different fraction of the datasets.

commonsense knowledge in its parameters so that it does not require much data for fine-tuning before obtaining a good performance. In contrast, the original T5 model requires much data for fine-tuning, which suggests it may fail to encode much commonsense knowledge and must fit the correlation patterns in the downstream datasets to get a good performance.

**Comparison of Generated Data** Table 5 shows the comparison of generated examples for the COMMONGEN test set between T5-base and CALM. We can see that the sentences generated by CALM are generally more acceptable in terms of commonsense plausibility while T5-base sometimes generates sentences that do not make sense.

| Concept-set | T5-base | CALM-base |
|---|---|---|
| Grass, Dog, Ball, Chase | a dog is chased by a ball on the grass. | dog chasing a ball in the grass. |
| Net, Cast, Boat, Water | fishing boat casts a net in the water. | fisherman casts a net into the water from a fishing boat. |
| Hole, Tree, Plant, Dig | a man digs a hole in a tree to plant a new tree . he digs the | man digging a hole to plant a tree. |
| Ingredient, Add, Pan, Fry | a pan filled with ingredients adds a touch of spice to the fry . | add the ingredients to a pan and fry. |
| Water, Hold, Hand, Walk | A man holding a hand and walking in the water. A man is holding water. | man holding a bottle of water in his hand as he walks down the street. |
| Place, Use, Metal tool | A man uses a metal tool to make a piece of metal. | woman uses a metal tool to make a piece of jewelry. |

Table 5: **Comparison of generated sentences with same concept-set.**

**Knowledge Probing** To investigate how much concept-centric knowledge our model pack, we conducted two probing methods with our model : Language Model Analysis (LAMA) probe (Petroni et al., 2019), Knowledge Intensive Language Task (KILT) (Petroni et al., 2020). We summarize the results on Table 6 and Appendix A.4. We could find that our model outperforms the baseline.

## 5 RELATED WORK

**Self-Supervised Language Representation Pre-Training.** Motivated by the fact that words can have different meanings in different contexts, contextual language representation methods (McCann

| Methods | MRR | Precision@50 | Precision@10 | Precision@1 |
|---|---|---|---|---|
| T5-Base | 11.53 | 38.52 | 21.60 | 5.93 |
| CALM (Mix-only) | 11.77 | 38.93 | 21.92 | 6.10 |
| CALM | **12.09** | **39.69** | **22.53** | **6.46** |

(a) LAMA probe

| Methods | FEVER | AY2 |
|---|---|---|
| T5-base | 76.65 | 74.97 |
| CALM (Mix-only) | 77.05 | 76.27 |
| CALM | **77.44** | **77.24** |

(b) KILT task

Table 6: **Experimental results on Knowledge Probing.** Left table shows the mean precision on LAMA probing task of ConceptNET. Right table shows the performance on Fact checking and Entity linking, which are from KILT task.

et al., 2017; Peters et al., 2018) have been developed and shown superior performance on downstream tasks compared with static word embeddings Mikolov et al. (2013); Pennington et al. (2014). More recently, large scale language models based on transformer architecture (Vaswani et al., 2017) pre-trained with either mask language modeling objective (Devlin et al., 2018; Liu et al., 2019; Lan et al., 2019) or mask span infilling objective (Lewis et al., 2019; Raffel et al., 2019) have been explored further advanced the state-of-the-art on multiple NLU and NLG tasks. Our method is based on these techniques and we focus on improving the commonsense reasoning ability of pre-trained text-to-text transformers. More recently, Clark et al. (2020) propose a new self-supervised pre-training objective called Replaced Token Detection (RTD). RTD uses a mask language model like BERT to fill in the mask and train a discriminator to predict whether a token is generated or real. This pre-training paradigm is related to our proposed joint training framework. Some major differences include that (1) Our method employs sentence-level distractors that are in general grammatically correct but not in line with commonsense, thus require the model to perform relational commonsense reasoning while RTD is a token-level discrimination task and can often be solved with syntactic and shallow semantic knowledge (Rosset et al., 2020); (2) Our method unifies generative and contrastive objectives with one model, which can be applied to both NLU and NLG downstream tasks; and (3) The discriminator in our framework is "contrastive", takes both the real sentence and the distractor as input simultaneously.

**Knowledge-augmented PTLMs.** As standard pre-trained language models usually do not explicitly model knowledge, a number of works have examined the problem of incorporating world knowledge with the PTLMs. Recent work Zhang et al. (2019); Peters et al. (2019); Wang et al. (2020); Liu et al. (2020) utilizes an external knowledge base to incorporate entity knowledge with PTLMs; however, these approaches require specialized resources like knowledge bases, which limits the domain they can be applied to. Xiong et al. (2020) proposes WikiLM that encodes world knowledge into the parameters of a BERT(Devlin et al., 2018)-like pre-trained model with a novel entity replacement detection objective that incorporates Wikipedia to form distractors. Their approach differs from ours because it requires an external knowledge base (i.e., Wikipedia) which limits the domain it can be applied, is limited to discriminative pre-training objectives and downstream tasks, and focuses on world knowledge instead of relational commonsense knowledge. More recently, (Rosset et al., 2020) propose KALM, an entity-aware language model with more world knowledge packed into its parameters. Their method is restricted to the training of language models instead of masked language models or text-to-text transformers which can be used for more downstream tasks. Also, all the aforementioned work mainly focuses on world knowledge of named entities. In contrast, our work mainly focuses on commonsense knowledge about quotidian concepts.

## 6 CONCLUSION

We propose novel self-supervised strategies that encourage the model to focus on concept-centric information that is related to commonsense understanding and reasoning instead of simple word co-ocurrence patterns so that the commonsense learning capability of pre-trained text-to-text transformers can be improved. Despite merely continually pre-trained on a small dataset with only around 20k steps, our CALM model consistently outperforms the T5-base model on all commonsense-related datasets, and even yields better performance compared with some larger size PTLMs on the COMMONGEN dataset. The performance gain is larger when we use fewer examples for fine-tuning on different downstream tasks, indicating that CALM effectively encodes more commonsense knowledge and rely less on fitting superficial patterns of datasets compared to traditional pre-trained language models. Our work suggests that text-to-text models can be pre-trained with better parameter and sample efficiency by carefully designed self-supervised objectives that focus more on the ability (e.g., commonsense reasoning ability) required by target tasks.

ACKNOWLEDGEMENT

This research is supported in part by the Office of the Director of National Intelligence (ODNI), Intelligence Advanced Research Projects Activity (IARPA), via Contract No. 2019-19051600007, the DARPA MCS program under Contract No. N660011924033 with the United States Office Of Naval Research, the Defense Advanced Research Projects Agency with award W911NF-19-20271, and NSF SMA 18-29268. We would like to thank all the collaborators in USC INK research lab for their constructive feedback on the work.

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

# A  APPENDIX

## A.1  PRE-TRAINING DETAILS

The following details apply to both base architecture and joint-training architecture. We implement our pre-train models using Pytorch-lightning (Falcon, 2019) and Hugginface's Pytorch Transformers (Wolf et al., 2019). For pre-training phase, we use the Adam optimizer with maximum sequence length 256, train batch size 8, gradient accumulation 8, warmup steps 10000, weight decay 0.01 and adam epsilon 1e-6. We train the models with 8 V100 GPUs and FP32 precision for 17 hours. The model is pre-trained for at most 3 epochs to prevent overfitting. We searched for the best learning rate for our model out of [1e-4, 2e-5, 2e-6, 5e-7].

## A.2  FINE-TUNING DETAILS

For fine-tuning, we use 4 V100 GPUs and use FP32. For all discriminative tasks, we use the Adam optimizer with maximum sequence length 256, batch size 4 and gradient accumulation 16. For generative task, we use the Adam optimizer with maximum source length 32, maximum target length 32, batch size 8, gradient accumulation 16. For all tasks, we use warmup fraction 0.01. Learning rates and train epochs are listed in Table 7.

| Hyperparameter | CommonsenseQA | OpenbookQA | PIQA | aNLI | CommonGEN |
|---|---|---|---|---|---|
| Learning rate | [1e-4, 2e-4, 3e-4] | [5e-5, 1e-4, 2e-4, 3e-4] | [1e-4, 2e-4, 3e-4] | [2e-5, 3e-5] | [2e-5] |
| Train Epochs | 20 | 20 | 20 | 10 | 20 |

Table 7: **Fine-tuning hyperparameters.**

## A.3  DATASET PROPERTIES

- **CommonsenseQA** (Talmor et al., 2018) is a multiple-choice question answering task, which picks the most appropriate answer on general commonsense questions.

- **OpenbookQA** (Mihaylov et al., 2018) is a multiple-choice question answering task, which is modeled after open book exams on elementary-level core science questions. The task requires open book fact and additional commonsense which is not contained in the book. To test the commonsense reasoning ability, we do not use open book fact.

- **PIQA** (Bisk et al., 2020) is multiple-choice question answering task, which chooses the most appropriate solution for physical commonsense questions.

- **aNLI** (Bhagavatula et al., 2019) is a binary-classification task, which picks the most plausible explanatory hypothesis given two observations from narrative contexts.

- **CommonGEN** (Lin et al., 2020) is a constrained text generation task, which generates a coherent sentence describing an everyday scenario using common concepts.

| Dataset | Train | Development | Test | Source Example | Target Example |
|---|---|---|---|---|---|
| CommonsenseQA | 9,741 | 1,221 | 1,140 | context: *What home entertainment equipment requires cable?* options: 1: *radio shack* 2: *substation* 3: *cabinet* 4: *television* 5: *desk* | 4 |
| OpenbookQA | 4,957 | 500 | 500 | context: *You can make a telescope with* options: 1: *straw* 2: *glass* 3: *candle* 4: *mailing tube* | 2 |
| PIQA | 16,113 | 1,838 | 3,084 | context: *When boiling butter, when it's ready, you can* options: 1: *Pour it onto a plate* 2: *Pour it into a jar* | 2 |
| aNLI | 169,654 | 1,532 | 3,040 | context: *It was my birthday. When I got home the party was set up for my brother.* options: 1: *I was so excited.* 2: *I was so mad.* | 2 |
| CommonGEN | 67,389 | 4,018 | 6,042 | generate a sentence with these concepts: *Apple Grow Tree* | *Apple grows* on the *tree* |

Table 8: **Properties of Commonsense benchmark datasets.**

## A.4 Knowledge Probing

LAMA probe is consisting of a set of knowledge sources, each comprised of a set of fact. It defines that a pre-trained language model knows a fact (subject, relation, object) such as (Bird, CapableOf, Fly) if it can predict masked objects in cloze statement such as "Birds can [MASK]". For evaluation, we first filtered out examples that mask label is not in vocabulary list of T5. Then, we evaluate the model based on how highly it ranks the ground truth token against every other word in a fixed vocabulary list of T5, and get mean precision at k to check whether the object is ranked among the top k results. We summarize the results of ConceptNet (Speer & Havasi, 2012) in Table 6. Unlike other language models which are optimised to masked word anywhere in a given sequence, T5 is trained with different denoising method. It might cause low performance on such slot filling task, but compared to T5, our model shows better performance compared to base model.

KILT task is a benchmark for assessing models that need to access specific knowledge in a defined snapshot of Wikipedia to solve tasks spanning five domains. The goal is to analyze the model whether it has task-agnostic representations of knowledge. We test our model on domain of fact checking, entity linking. Fact checking verifies textual claims against textual sources. For this task, we use FEVER (Thorne et al., 2018) which is a large dataset for claim veracity that requires evidence from multiple Wikipedia pages to determine whether the claim is supported or refuted. Entity Linking assigns Wikipedia page to entities mentioned in text. We use AIDA CoNLL-YAGO (AY2) (Hoffart et al., 2011) which supplements the CoNLL 2003 (Tjong Kim Sang & De Meulder, 2003) with Wikipedia URL annotations for all entities.

## A.5 Experiments with BART as Backbone

To show that our approach is versatile to different pre-trained models, we conduct experiments with BART as the backbone model. We can see that our approach consistently and significantly (with p-value < 0.01) improves BART-base on all datasets. This result shows that our method is versatile to different pre-trained models.

| Methods | CSQA | OBQA | PIQA | aNLI | CommonGEN | | | |
|---|---|---|---|---|---|---|---|---|
| | Accuracy (official dev) | | | | BLEU-4 | METEOR | CIDEr | SPICE |
| BART-base (Mix-only) | 56.31($\pm$0.28) | 58.30($\pm$1.1) | 67.53($\pm$1.01) | 59.85($\pm$1.14) | 25.10 | 29.50 | 13.16 | 30.20 |
| CALM (BART-base) | **58.22($\pm$0.21)** | **59.10($\pm$1.0)** | **69.40($\pm$1.23)** | **61.28($\pm$0.30)** | **26.40** | **29.90** | **13.71** | **31.10** |

Table 9: **Experimental results with BART as backbone model.** Best models are bold.

## A.6 Experiments with Noun/Verb as Concepts

We also conducted an ablation study about the choice of using either nouns or verbs as concepts. We can see that using either nouns-only or verbs-only as concepts for our approach leads to substantial performance drop. This supports our choice about using both nouns and verbs as concepts.

| Methods | CSQA | OBQA | PIQA | aNLI | CommonGEN | | | |
|---|---|---|---|---|---|---|---|---|
| | Accuracy (official dev) | | | | BLEU-4 | METEOR | CIDEr | SPICE |
| CALM | **63.32($\pm$0.35)** | **60.90($\pm$0.4)** | **71.01($\pm$0.61)** | **63.20($\pm$0.52)** | **26.40** | **31.40** | **13.88** | **33.00** |
| CALM-nouns | 62.45($\pm$0.42) | 59.40($\pm$0.5) | 69.05($\pm$0.70) | 61.55($\pm$0.58) | 25.70 | 31.20 | 13.17 | 32.60 |
| CALM-verbs | 62.51($\pm$0.47) | 59.10($\pm$0.7) | 69.24($\pm$0.65) | 61.40($\pm$0.51) | 25.60 | 31.20 | 13.24 | 32.60 |

Table 10: **Experimental results with Noun/Verb as Concepts.** Best models are bold.

## A.7 Human Evaluation on CommonGEN generations

We conducted a human evaluation of CommonGEN predictions between T5 and CALM. We asked three annotators to choose the most reasonable sentence between T5-base and CALM-base predictions. The evaluation was conducted on 50 test sentences in binary selection by majority voting. Cohen's Kappa score, which is a measurement of inter-annotator agreement, was 0.73. Annotators say that for 60% of test sentences, CALM-base generated better.

