# OpenReview forum: "Pre-training Text-to-Text Transformers for Concept-centric Common Sense"
_ICLR.cc/2021/Conference — ICLR 2021 Poster_

### Official Review · AnonReviewer1 · 2020-10-24
**Nice approach to incoporate commonsense into large models but needs more evaluation**

**Rating:** 8
**Confidence:** 4

**Review:**

#### Summary
This paper addresses the issue of incorporating commonsense into large pretrained language models. They propose a pretraining method that does not leverage knowledge graphs but rather use approaches which involve corrupting the input sentence in various ways to recover the original sentence. This methodology is used to try to bake in commonsense into models. Results are shown on both discriminative and generative common sense datasets

#### Novelty and clarity
The training procedure of corrupting inputs to retrieve outputs is not new but the use on commonsense tasks does seem novel and also is an interesting approach. The paper was very clear to read and the technical aspects were well described.

#### Strengths
(1) The use of a self-supervised approach is great because it requires no annotation and the training procedure is simple. It is also described well
(2) The variety of baselines used is good and comparison against models larger than the proposed model is interesting to see.
(3) The use of generated sentences to improve language models on hard areas like commonsense and offensiveness is a great idea as it can help make the model more robust


#### Weaknesses
(1) There should be a more comprehensive set of results completed to see how much improvement this model has. Mainly on the CommonGen class there should be some manual evaluation done similar in the original paper to see if the outputted sentences make sense and that the improvement in automatic metrics is carried over into human evaluation.
(2) A key aspect to look into is the robustness of this model. In the C2S approach the concepts were shuffled to generate the correct sentence. During inference time if the concepts were shuffled in a different manner would the model still be able to generate the correct sentences? There was three random seeds used but as was said "the performance is sensitive to different random seeds." which seems that the model isn't as robust to newly seen inputs

#### Detailed Comments
If spacy was also used for POS tagging along with tokenization this should be made clear. Also for every sentence was there 3 nouns/verbs extracted?
One thing I'm unclear about is in Table 2 why is the "Our T5-Base" better than the "T5-Base" above? Is this T5 with additional epochs? I think this should be made clear. Additionally I wouldn't say and "is only slightly worse than KG-BART." It seems a lot worse especially on BLEU and CIDER. It is nice to see a smaller model is beating a larger model on some metrics
"The difference between CALM and CALM (Joint) is that the former is initialized by the CALM(Mix)." Did you mean to say latter instead of former? Also I don't see CALM (Join) in the table. I'm assuming this is CALM without Mix warmup


#### Questions for the Authors
1) How did you ensure shuffling the sentences still has grammatical correctness? A sentence like "Running I am" is not grammatically correct.
2) Instead of a POS tagger why did you not use an NER extractor? Also wouldn't swapping different fruits into sentences like replacing "Apples grow on trees" with "Watermelons grow on trees" help with robustness
3) And what point is the generative model good enough that it doesn't help to create distractor sentences

---

> ### Author Response · Authors · 2020-11-15
> **Response to Reviewer 1 (1/2)**
>
> Thanks for your supportive review and valuable feedback!
>
> ### Response to weakness:
>
> >Weakness 1:
> >
> >There should be a more comprehensive set of results completed to see how much improvement this model has. Mainly on the CommonGen class there should be some manual evaluation done similar in the original paper to see if the outputted sentences make sense and that the improvement in automatic metrics is carried over into human evaluation.
>
> This is a good suggestion! To make the empirical results more convincing, we have conducted a set of additional experiments. Specifically, we have investigated whether our CALM model packs more concept-centric knowledge compared to the original T5-base, we conducted two well-recognized knowledge probing analysis of compared models: Language Model Analysis (LAMA) probe (Petroni et al.,2019) and Knowledge Intensive Language Task (KILT) (Petroni et al., 2020). The results are as follows (presented in Table 6 in the updated draft):
>
> - LAMA Probe
> |method    | MRR  | Precision@50| Precision@10| Precision@1|
> | ----------- | :-------: | :--------: | :-------: | :-------: |
> |T5-base   |11.53  |  38.52             |     21.60         |         5.93      |
> |CALM	    | **12.09** |  **39.69**        |     **22.53**         |        **6.46**       |
>
> - KILT tasks (acc)
> |method | FEVER (fact verification )| AY2 (entity linking) |
> | ----------- | :-------: | :--------: |
> |T5-base|           76.65                     | 74.97                      |
> |CALM   |           **77.44**                     | **77.24**                    |
>
> We can see CALM consistently outperforms the T5-base model across a set of knowledge-related tasks.
>
> We have also conducted experiments with T5-large as a larger backbone model and find our method still improves the T5-large model by a consistent margin compared to that with the base-size backbone (CALM-large vs T5-large in Table 2):
>
> |method            | CSQA | OBQA  |   PIQA  | aNLI   |  CommonGEN                   |
> | ----------- | :-------: | :--------: | :-------: | :-------: | :-------: |
> |T5-large           | 69.81  |  61.40  |  72.19   | 75.54 |  25.10/31.90/12.54/32.90 |
> |CALM-large	  | **70.71**  |  **66.00**  |  **74.59**   | **76.96** |  **26.70**/**32.00**/**14.01**/**33.30** |
>
> As for the CommonGen dataset, we’ve included some qualitative examples in the draft that show sentences generated by CALM make more sense than that generated by T5-base. We’re currently doing a human evaluation per your suggestion and will circle back in two or three days.
>
> **New results for Human Evaluation:**
> We conducted a human evaluation of CommonGEN predictions.
> We asked three annotators to choose the most reasonable sentence between T5-base and CALM-base predictions.
> Here's the detail:
> - Number of test sentences: 50
> - Binary selection by majority voting
> - Kappa score (inter-annotator agreement) : 0.73
> - CALM-base : 0.6 / T5-base : 0.4
>
> We can find that CALM predictions usually more reasonable than T5.
>
> >Weakness 2:
> >
> >A key aspect to look into is the robustness of this model. In the C2S approach the concepts were shuffled to generate the correct sentence. During inference time if the concepts were shuffled in a different manner would the model still be able to generate the correct sentences? There was three random seeds used but as was said "the performance is sensitive to different random seeds." which seems that the model isn't as robust to newly seen inputs.
>
> Since the concepts are shuffled randomly and the corpus for pre-training is relatively large, we believe it will be robust to different orders for shuffling. As for the sensitivities of performance, it is not because the input is shuffled differently, instead, we suspect it is because the training sets of some datasets are small, which corresponds to the findings in (Dodge et al. 2020).
>
> ==============================\
> Reference: \
> Petroni et al.,2019 Language Models as Knowledge Bases? EMNLP 2019\
> Petroni et al.,2020 Kilt: a benchmark for knowledge intensive language tasks arxiv\
> Dodge et al. 2020 Fine-Tuning Pretrained Language Models: Weight Initializations, Data Orders, and Early Stopping

---

> > ### Author Response · Authors · 2020-11-15
> > **Response to Reviewer 1 (2/2)**
> >
> > ### Response to detailed comments:
> >
> > - We’ve made it clear that we used Spacy for both tokenization and POS tagging.
> > - The T5-base (our implementation) uses different hyperparameter settings that suit better to base-size models compared to that - reported in the leaderboard.
> > - We have corrected the claim about performance comparison with KG-BART.
> > - We have made it clear by using the terms CALM(w/o mix warmup) thoroughly.
> >
> > ### Response to questions:
> >
> > >Q1:
> > >
> > >How did you ensure shuffling the sentences still has grammatical correctness? A sentence like "Running I am" is not grammatically correct.
> >
> > As described in the paper, we only shuffle concepts between the same category (i.e. shuffle nouns with nouns and verbs with verbs), so if the sentence is “apple grows on the tree”, the corrupted sentence will be “tree grows on the apple” rather than “grows tree on the apple”. This ensures most shuffled sentences grammatically correct.
> >
> > >Q2:
> > >
> > >Instead of a POS tagger why did you not use an NER extractor? Also wouldn't swapping different fruits into sentences like replacing "Apples grow on trees" with "Watermelons grow on trees" help with robustness
> >
> > That’s a good point! Actually, most entities can be extracted by POS tagging from the PROPNOUN tag. Therefore we did not use NER extractor since we do not need to know whether the entities are names or places. This saves some computation. Replacing concepts with noisy concepts that are similar to the original one is a good point for introducing extra noise signals. We are currently running experiments to verify this and will post the results during discussion or update them in the final version if it has not been done within the discussion period.
> >
> > >Q3:
> > >
> > >And what point is the generative model good enough that it doesn't help to create distractor sentences
> >
> > In the joint training framework, the model exploits the knowledge that is already packed in its parameters to generate sentences. Even if the output sentences are of good quality, they may still differ from real sentences, and training the discriminator to distinguish them forces the model to acquire new commonsense knowledge. Therefore, the model is trained to iteratively improve upon itself in a self-play fashion.  In addition, generating distractors for contrastive learning in a GAN-style is related to previous work for language model pre-training like ELECTRA, which uses a masked language model to generate distractor tokens. This choice makes the distractor hard to distinguish and makes the training more informative.

---

### Official Review · AnonReviewer2 · 2020-10-28
**Interesting work**

**Rating:** 8
**Confidence:** 4

**Review:**

Summary:
This paper proposes the “Concept Aware Language Model” (CALM) --- a pre-trained T5 Transformer is trained on self-supervised intermediate tasks to learn relational commonsense knowledge, before fine-tuning on downstream tasks. The intermediate tasks include (1) concept to sentence (c2s) generation - given a list of permuted concepts (verbs and nouns), generate the target sequence, (2) concept order recovery (cor) - given a sequence with the order of concepts (verbs and nouns) shuffled, generate the original sequence, (3) given a sequence and it’s perturbed version (concepts shuffled), generate the sequence (classification but as a generation task). Training is first done in a multi-task fashion, followed by a stage of training that uses generated outputs from (1) and (2) for (3).

The goal of the paper is to show that carefully designed objectives for self-supervised intermediate task training add relational commonsense knowledge which helps improve model performance on downstream commonsense reasoning tasks.

Strengths:
- S1 - The idea of self-supervised intermediate task training is an exciting one especially in the form of adding reasoning capabilities that BERT, GPT, T5 etc. might not be acquiring during the large-scale pre-training phase.
- S2 -  As shown by results in Table 1 on 5 commonsense reasoning tasks, the intermediate task training proposed in this work improves performance over and above the T5 model (and variants including with salient span masking for concepts).
- S3 - The experiments involved averaging across 3 random seeds and the authors have reported confidence intervals.

Weaknesses and questions:
- W1 - One missing baseline is T5 trained to unshuffle entire sequences - given an input with the tokens shuffled, generate the original sequence. This would show how much value c2s and cor are really adding. The current T5 baselines are all trained purely for infilling which seems a bit unfair compared to CALM which is generating entire sequences.
- W2 - Given a T5 model that can score sequences (maybe after training on the autoencoding objective), would it score “apples grow on trees” higher than “trees grow on apples”? If yes, then the model seems to already exhibit this style of reasoning. Would it score “apples grow on trees” and “apples grow in the ground” similarly? The distinction here is between sequences that are non-grammatical or unlikely to ever appear versus sequences that may have appeared (eg: “potatoes grow in the ground”). Presently, (a) it’s unclear if the designed objectives are providing commonsense reasoning above something the model can know from autoregressive language model scoring, and (b) it appears that the objectives are not designed to add relational commonsense knowledge of the sort where we know apples don’t grow in the ground.
- W3 - c2s is designed specifically to do well on CommonGen. Are the gains on this task smaller than what you might have expected? If yes, why isn’t it helping more?
- W4 - Figure 4 needs error bars, the dev sets are really small and it’s hard to interpret which differences are significant.



__UPDATE__

Thanks for the incredibly detailed response! I've raised my score to a 8.
I do in general quite like the paper, and the responses here are thought-provoking. I'm not sure I'm totally convinced by the WSC results comparing CALM the classifier to T5 the sequence scorer. Not sure if it's an apples to apples comparison...but I'm not sure there's a straightforward setup for this, and perhaps it starts to get beyond the scope of what's being presented here.

---

> ### Author Response · Authors · 2020-11-15
> **Response to Reviewer 2**
>
> Thanks for your supportive review and valuable feedback!
>
> >Weakness 1:
> >
> >One missing baseline is T5 trained to unshuffle entire sequences - given input with the tokens shuffled, generate the original sequence. This would show how much value c2s and cor are really adding. The current T5 baselines are all trained purely for infilling which seems a bit unfair compared to CALM which is generating entire sequences.
>
> Training a text-to-text transformer with all tokens in the input shuffled and generating the original sentence is explored in the original T5 paper as a variant and the performance is significantly worse (see the Deshuffling variant in Table 4 in https://arxiv.org/pdf/1910.10683.pdf). We suspect this is because the task may be too difficult to train. We also conducted a variant of fine-tuning T5-base with this objective. The results are shown below:
>
> |method     | CSQA | OBQA  |   PIQA  | aNLI   |  CommonGEN                   |
> | ----------- | :-------: | :--------: | :-------: | :-------: | :-------: |
> |T5-base       | 61.88  |  58.20  |  68.14   | 61.10 |  24.90/31.20/12.99/32.40 |
> |T5-base + deshuffling | 61.32  |  57.40  |  67.75   | 60.70 |  25.10/31.20/13.21/32.50 |
> |CALM | **63.32**  |  **60.90**  |  **71.01**   | **63.20** |  **26.40**/**31.40**/**13.88**/**33.00** |
>
> We can see that the performance after fine-tuning sentence de-shuffling is lower than the original T5 except for CommonGEN which is similar to this task and performs significantly worse than CALM on all datasets. This confirms the effectiveness of our concept-centric objectives.
>
> >Weakness 2:
> >
> >Given a T5 model that can score sequences (maybe after training on the autoencoding objective), would it score “apples grow on trees” higher than “trees grow on apples”? If yes, then the model seems to already exhibit this style of reasoning. Would it score “apples grow on trees” and “apples grow in the ground” similarly? The distinction here is between sequences that are non-grammatical or unlikely to ever appear versus sequences that may have appeared (eg: “potatoes grow in the ground”). Presently, (a) it’s unclear if the designed objectives are providing commonsense reasoning above something the model can know from autoregressive language model scoring, and (b) it appears that the objectives are not designed to add relational commonsense knowledge of the sort where we know apples don’t grow in the ground.
>
> Great points! For the first question, according to (Trinh et al. 2018), a language model can help solve the Winograd Schema Challenge dataset and therefore possess this kind of knowledge, and that’s why pre-training can improve the performance on commonsense-related datasets. However, as stated in the paper, these kinds of knowledge are implicitly learned during traditional pre-training and our method encourages the model to learn this knowledge more effectively. To further demonstrate this, we have conducted an experiment where we use CALM and T5-base to solve the Winograd Schema Challenge in an unsupervised setting by using the discrimination objective (prefix) of CALM to distinguish which sentence is correct and use T5-base to score the sentences respectively. We find that CALM outperforms T5-base by over 3 points on WSC, demonstrating our objectives provide more commonsense knowledge for the model.
>
> For the second question, sentences in the training data are typically longer than the example and there may be many distractor concepts like the ground in the input concept-set. In this scenario, the model is encouraged to learn this kind of knowledge. Also, as suggested by Reviewer 1,  replacing concepts with noisy concepts that are similar to the original one is a great point for introducing extra noise signals. We are currently running experiments to verify this and will post the results during discussion or update them in the final version if it has not been done within the discussion period.
>
> >Weakness 3:
> >
> >c2s is designed specifically to do well on CommonGen. Are the gains on this task smaller than what you might have expected? If yes, why isn’t it helping more?
>
> Actually, the performance gain of CALM upon T5-base is more significant than it looks like. For BLEU and CIDEr, the improvement of CALM upon T5-base is almost half of that yielded by T5-large, which has nearly 4x parameters. Also, a possible reason is that CALM is trained in a multi-task fashion instead of C2S only.
>
> >Weakness 4:
> >
> >Figure 4 needs error bars, the dev sets are really small and it’s hard to interpret which differences are significant.
>
> The results in Figure 4 are not on dev sets but on test sets, which are larger. We’ll add the error bar to make it more convincing. Thanks for your suggestion.

---

> > ### Author Response · Authors · 2020-11-23
> > **Reminder for the discussion**
> >
> > Dear Reviewer 2,
> >
> > We want to send you a friendly reminder for the discussion.
> > Here are the things that we have added and resolved by your valuable feedback!
> >
> > - We show results of **deshuffling**. The result shows that performance after fine-tuning sentence de-shuffling is lower than original T5 except for CommonGEN.
> > - We add discussion of our designed objectives in terms of commonsense knowledge.
> > - We add discussion of CommonGen results. We argue that the performance gain of CALM upon T5-base is more significant than it looks like.
> >
> > Moreover, we added additional experiments of applying our methods into large models and other pre-trained language models, and also more ablation studies.
> >
> > We thank you again for your valuable comments, and we would appreciate it if you could review our improvement and give us another round of feedback.
> >
> > Thanks.

---

> > ### Author Response · Authors · 2020-11-24
> > **Response to Reviewer 2 Cont**
> >
> > >Weakness 2:
> > >
> > >Given a T5 model that can score sequences (maybe after training on the autoencoding objective), would it score “apples grow on trees” higher than “trees grow on apples”? If yes, then the model seems to already exhibit this style of reasoning. Would it score “apples grow on trees” and “apples grow in the ground” similarly? The distinction here is between sequences that are non-grammatical or unlikely to ever appear versus sequences that may have appeared (eg: “potatoes grow in the ground”). Presently, (a) it’s unclear if the designed objectives are providing commonsense reasoning above something the model can know from autoregressive language model scoring, and (b) it appears that the objectives are not designed to add relational commonsense knowledge of the sort where we know apples don’t grow in the ground.
> >
> > In addition to the previous response, we also conducted an experiment using T5-base and CALM-base to identify original vs. concept-perturbed sentences. We find that CALM-base achieves over 5% accuracy improvement upon CALM-base. This confirms that, while our proposed objectives are not designed to introduce “add new relational knowledge” to the model — since we’re only perturbing existing sentences, they make a pre-trained acquired relational commonsense knowledge in texts more effectively.
> >
> > As for concept set generalization, we appreciate this idea and will leave it as future work.
> >
> > >Weakness 3:
> > >
> > >c2s is designed specifically to do well on CommonGen. Are the gains on this task smaller than what you might have expected? If yes, why isn’t it helping more?
> >
> > - Additional Response:
> > In addition, we have conducted experiments with larger backbone models and the results on CommonGEN are as follows:
> >
> >  Results on CommonGEN
> > |method                                 | # parameters    |  BLEU4|  METEOR | CIDEr | SPICE |
> > | ---------------------------   |   :-------:   |   :--------:   | :-------: | :-------: | :-------: |
> > |BART                                      |  406M                 |  26.30  |  30.90       | 13.92  | 30.60 |
> > |T5-large                                 |   774M                |  28.60  |  30.10       | 14.96  | 31.60 |
> > |CALM-large	                       |   774M                |  29.50  |  31.90       | 15.61  | **33.20** |
> > |KG-BART (SOTA, with KG)   |  406M                 |  **30.90**  |  **32.40**       | **16.83**  | 32.70 |
> >
> > We can see that CALM-large consistently outperforms T5-large by a large margin. It also achieves state-of-the-art performance on the SPICE metric, which is the most important metric for CommonGEN because it correlates best with human evaluation according to the original paper of CommonGEN.
> >
> > We have also conducted a human evaluation of CommonGEN predictions.
> > We asked three annotators to choose the most reasonable sentence between T5-base and CALM-base predictions.
> > Here's the detail:
> > - Number of test sentences: 50
> > - Binary selection by majority voting
> > - Kappa score (inter-annotator agreement) : 0.73
> > - CALM-base : 0.6 / T5-base : 0.4
> >
> > We can find that CALM predictions usually more reasonable than T5.

---

### Official Review · AnonReviewer3 · 2020-10-28
**Interesting Proposed Objectives, Weak Results**

**Rating:** 7
**Confidence:** 4

**Review:**

This paper suggestsan intermediate training regime that can be used between pretraining and the end-task finetuning. The authors suggest that their method captures more commonsense knowledge by being focused on capturing knowledge about “concepts”. Four different denoising objectives—two generative and two discriminative—are proposed and described in detail, with various possible ways of optimizing for all four. Experimental results show improvements over both the base T5 model and the large T5 model. The proposed method achieves SoTA results on CommonGen with slightly more than half the parameters of the current SoTA model.  Ablations show the necessity of applying a 2-step intermediate training scheme with mixed training followed by joint training.  CALM shows better results with less data than the base model.

Strengths:
- Unifying generative and contrastive training is an important and interesting goal.
- The objectives suggested are cheap to compute and seem to increase the signal available in the data.
- Extensive results show improvements over a base model and a larger model across a range of tasks.
- Performance with relatively little finetuning data are encouraging.

Weaknesses:
- Somewhat weaker results on some CommonGen metrics are disappointing.
- Using “concept” to stand in for verbs and nouns is somewhat confusing.

After reading other reviews and the authors’ responses to all of the reviewers, I recommend this paper by accepted—extensive results show that the CALM objectives offer more signal from data than current pretraining methods.

This paper suggests a number of cheap-to-compute corruptions of the input data that, when used to reconstruct the input, enrich the underlying model. These objectives certainly improve over the original T5 base _and_ larges models that are used as initializations, and especially outperform the base model in the low-data regime. The authors use objectives which capture both generative and discriminative information, which some have suggested contain mutually beneficial signal but have not been unified in a single training method.

Below are two paragraphs from my original review. The authors have done further experiments and show that there are still gains on these tasks when model sized is increased significantly. Furthermore, they have clarified that on the key metric of CommonGen they achieved SoTA with only slightly more than half the parameters of the current SoTA model. I therefore believe that these results show merit.
> However, in every task except CommonGEN the authors do not discuss any methods that are even close to the state of the art. For CSQA, the best number in this paper is 63.32 vs. 79.5 on the current leaderboard. Similar numbers are true for the rest of the tasks: 60.90 vs. 87 for OBQA, 71.01 vs. 90 for PIQA, and 63.20 vs.  89.70 for aNLI. I do not believe that SoTA results are necessary to write a good paper, and indeed the obsession our field has with SoTA is unhealthy. Yet, it is difficult for me to trust that the effects in this paper will generalize to better performing models without further evidence: what if the CALM intermediate objectives only help with mistakes that larger models do not make in the first place?

> On the generative task CALM performs closer to SOTA, but it improves only slightly on T5. This is especially disappointing as the objectives introduced _directly_ match the task in CommonGEN, making this intermediate training a form of noisy training data rather than pretraining.

I still feel that the authors’ use of “concept” and “commonsense” is vague, when their method can be defined more clearly with more mundane terminology. In practice, the authors use nouns and verbs as their concepts, which is fine in terms of pretraining objectives, but surely does not capture the generality of concepts. The authors have somewhat clarified in this in their updated version.

Finally, the CALM intermediate objectives share many properties with all of the datasets tested on and are likely calibrating the model to the kind of correlations they should expect to predict in advance of finetuning. One way this can be seen is  that the slopes of the T5 and CALM lines are very similar after an initial “bump” which T5 likely needs to calibrate to the new distribution. This makes claims of learning “commonsense” hard to verify, though I do agree that _something_ relevant to solving these problems is clearly being learned.

Altogether, I think this paper makes an interesting contribution to the question of: How can we get the most pretraining signal from unstructured data using off-the-shelf tools? I recommend this paper for acceptance, though I encourage the authors to revise their paper to make this the focus of the story, rather than the vaguely defined notion of “concept”.

---

> ### Author Response · Authors · 2020-11-15
> **Response to Reviewer 3 (1/2)**
>
> Thanks for your insightful review and valuable feedback!
>
> >Weakness 1:
> >
> >The scores reported by the base & proposed models are so far below SoTA it is unclear where these techniques will generalize to models that are likely to be used.
>
> Thank you for the comment! We admit that the performance of base-size CALM and T5 are far from SOTA numbers. However, we believe that the performance comparison of base size pre-trained models is meaningful because for SOTA models that contains 11B parameters:(1) the computational cost for continually pre-training them with our method is very computationally expensive, and (2) they are unlikely to be used in real-world applications because T5-3B and T5-11B model can’t be fitted in the GPU memory for inference even batch size set as 1 and the inference latency would be unsatisfactory. This is also suggested by the fact that the most popular (in terms of download times at huggingface’s model hub) pre-trained models are mostly base size models.
>
> To test whether our proposed method can generalize to larger models, we continually pre-trained CALM-large with the T5-large model as the backbone after the submission deadline. The result comparison between CALM-large and T5-large is shown as follows (also see Table 2 in the revised draft).
>
> - Results of discriminative tasks
> |     method                  | #parameters      |    CSQA    |    OBQA   |   PIQA   |    aNLI    |
> | ---------------------------   |   :-------:   |   :--------:   | :-------: | :-------: | :-------: |
> |     T5-large                 |      774M                | 69.81  |  61.40  |  72.19   | 75.54 |
> |     CALM-large	       |      774M                | 71.31  |  66.00  |  75.11   | 77.12 |
> |     BERT-large            |      345M                | 57.06  |  60.04  |  67.08   | 66.75 |
> |     RoBERTa-large     |      345M                | 71.81  |  63.90  |  76.90   | 82.35 |
> |     SOTA                      |      11B                   | 79.1    |  87.2    |  90.13   | 89.70 |
>
> - Results on CommonGEN
> |method                                 | # parameters    |  BLEU4|  METEOR | CIDEr | SPICE |
> | ---------------------------   |   :-------:   |   :--------:   | :-------: | :-------: | :-------: |
> |BART                                      |  406M                 |  26.30  |  30.90       | 13.92  | 30.60 |
> |T5-large                                 |   774M                |  28.60  |  30.10       | 14.96  | 31.60 |
> |CALM-large	                       |   774M                |  29.50  |  31.90       | 15.61  | **33.20** |
> |KG-BART (SOTA, with KG)   |  406M                 |  **30.90**  |  **32.40**       | **16.83**  | 32.70 |
>
> We can see that **CALM-large consistently improves upon T5-large on all tested datasets (statistically significantly with p-value < 0.01)**. The improvement is consistent with the improvement of CALM-base upon T5-base. This shows that our method can hopefully generalize well to even larger models like T5-3B and T5-11B.
>
> We have also tested the effectiveness of CALM on other tasks and datasets to make the empirical results more convincing. Specifically, we have investigated whether our CALM model packs more concept-centric knowledge compared to the original T5-base, we conducted two well-recognized knowledge probing analysis of compared models: Language Model Analysis (LAMA) probe (Petroni et al.,2019) and Knowledge Intensive Language Task (KILT) (Petroni et al., 2020). The results are as follows (presented in Table 6 in the updated draft):
>
> - LAMA Probe
> |method    | MRR  | Precision@50| Precision@10| Precision@1|
> | ----------- | :-------: | :--------: | :-------: | :-------: |
> |T5-base   |11.53  |  38.52             |     21.60         |         5.93      |
> |CALM	    | **12.09** |  **39.69**        |     **22.53**         |        **6.46**       |
>
> - KILT tasks (acc)
> |method | FEVER (fact verification )| AY2 (entity linking) |
> | ----------- | :-------: | :--------: |
> |T5-base|           76.65                     | 74.97                      |
> |CALM   |           **77.44**                     | **77.24**                    |
>
> We can see CALM consistently outperforms the T5-base model across a set of knowledge-related tasks.

---

> > ### Author Response · Authors · 2020-11-15
> > **Response to Reviewer 3 (2/2)**
> >
> > >Weakness 2:
> > >
> > >The objectives used are essentially reframings of the rules used to generate CommonGEN, and yet scores on CommonGEN are not very impressive.
> >
> > First, only the C2S objective is similar to that used to generate CommonGEN while the other objectives, including COR, the contrastive objective, and the joint-training framework are different from the rules for constructing CommonGEN and their usefulness is confirmed through ablation study and different versions of the CALM model. Second, actually, the performance gain of CALM upon T5-base is more significant than it looks like. Specifically, CALM achieves state-of-the-art performance on the SPICE metric, which is the most important metric for CommonGEN because it correlates best with human evaluation according to the original paper of CommonGEN. For BLEU and CIDEr, the improvement of CALM upon T5-base is almost half of that yielded by T5-large, which has nearly 4x parameters.  In addition, experiments with larger backbone models show that CALM-large consistently outperforms T5-large by a large margin.
> >
> > In addition, we conducted a human evaluation of CommonGEN predictions.
> > We asked three annotators to choose the most reasonable sentence between T5-base and CALM-base predictions.
> > Here's the detail:
> > - Number of test sentences: 50
> > - Binary selection by majority voting
> > - Kappa score (inter-annotator agreement) : 0.73
> > - CALM-base : 0.6 / T5-base : 0.4
> >
> > We can find that CALM predictions usually more reasonable than T5.
> >
> > >Weakness 3:
> > >
> > >The notion of “concepts” remains vague throughout the paper as do claims regarding commonsense.
> >
> > We have refined most of our claims in the revised version per your suggestion. Specifically, we defined the concepts to be nouns and verbs extracted by Spacy POS tagger. And our claim about commonsense is that our training objective encourages relational and compositional commonsense reasoning and help pack more commonsense knowledge into the parameter of a pre-trained model.
> >
> > >Concerns:
> > >
> > >The CALM intermediate objectives share many properties with all of these datasets and are actually calibrating the model to the kind of correlations they should expect to predict in advance of fine-tuning. One way this can be seen is that the slopes of the T5 and CALM lines are very similar after an initial “bump” which T5 likely needs to calibrate to the new distribution.
> >
> > It is true that CALM intermediate objectives share many properties with CommonGEN. For other discriminative tasks, the shared properties is that both of them require the model to reason with concepts, and this is an essential part of commonsense reasoning. The learning curve shows that with CALM intermediate objectives, the model needs less training data to achieve a good performance and suggests CALM may encode more commonsense knowledge in advance. It is common that the slope of different methods with more data are similar for data size-performance curves because the performance tends to saturate with more data.
> >
> > >Question:
> > >
> > >Are nouns and verbs distinguished between in the list of words given to the model? On page 3 it’s implied that verbs come first, but is there a delimiter between the verb list and the noun list?
> >
> > In C2S, Nouns and Verbs are not distinguished and jointly shuffled, as shown by the permute function in Eq (1). In COR, To ensure grammatical correctness of shuffled sentences, Nouns and Verbs are distinguished because nouns are only shuffled with nouns and vice versa, as indicated by the concept-permute function. The notation on Page 3 is just for simplicity and there’s no delimiter between the verb list and the noun list. We’ve made it clear in the revised version.

---

> > > ### Author Response · Authors · 2020-11-20
> > > **Look Forward to Hearing From You**
> > >
> > > Dear Reviewer 3,
> > >
> > > We appreciate your valuable feedback and comments. We could improve our paper in a good way with your great feedback!!
> > >
> > > We want to send you a friendly reminder that the first phase of the response period is completed, and want to hear back from you about our response.
> > >
> > > If you could let us know any other concerns that we could not address in the response, we'd be happy to talk about it! Thanks again for your time, and your response is eagerly awaited.
> > >
> > > Thanks.

---

> > > > ### Comment · AnonReviewer3 · 2020-11-25
> > > > **Very Informative Rebuttal!**
> > > >
> > > > Thank you for the detailed response and large number of new reported experiments!
> > > >
> > > > I have raised my score to 7, because I believe your  new experiments show encouraging results as model size grows and you have clarified some of my misunderstanding in your response. I discuss each issue below:
> > > >
> > > > > New Results
> > > >
> > > > I agree that the your results with the addition of larger models and more datasets show that CALM has a significant, if somewhat moderate effect.
> > > >
> > > > > CommonGEN Performance
> > > >
> > > > Thank you for clarifying what you think the gains are here—I agree that CALM is significantly more parameter efficient. Re-reading the CommonGen paper I agree that SPICE is the most important metric and also that your reimplementation of T5 far outperforms theirs, which makes this more impressive. Thanks for walking me through this!
> > > >
> > > > > “Concepts” as a Term
> > > >
> > > > I am still critical of your use of the term “concept”—but I do appreciate your updates and think this is more in line with general definitions of the community. My personal view is that “concept” is a very vague idea so to use it as motivation might be fine, but using it as a term here is a bit of an over-claim. That said, I do think this falls within the variation of usage in the community.
> > > >
> > > > > Data Effectiveness / Specificity of Training Regime
> > > >
> > > > I am still unconvinced that we have a meaningful way to distinguish between the setup of the CALM objectives and the datasets that NLU benchmarks tend to use—but solving this problem is out of the scope of this paper. I do not think that the data curve as shown makes it clear whether the CALM objective essentially calibrated the models for the dataset in advance of finetuning or encodes “deeper knowledge”. I would warn against making too many claims about the knowledge in a model, which is still very much a blackbox, but I don’t think not being able to show this deeper knowledge is present can be considered a fault of this paper.
> > > >
> > > > > Nouns & Verbs
> > > >
> > > > Thank you for clarifying this in the update!
> > > >
> > > > All in all, I now much more convinced that CALM adds information over current pre-training regimes. As I mentioned at the beginning of this response, I have raised my score for this paper to a 7 and recommend acceptance!

---

### Official Review · AnonReviewer4 · 2020-11-01
**The paper suggests an interesting redirection for commonsense reasoning, but lacks of rigorous experimentation to justify the contributions.**

**Rating:** 4
**Confidence:** 4

**Review:**

This paper proposes two self-supervised pre-training tasks to further pre-train a pre-trained language model for commonsense reasoning. The first task is called concept-to-sentence generation, which reconstructs input sentences from noun/verb phrases extracted from the input. The second task is called concept order recovering, which predicts original sentences after shuffling the order of noun/verb phrases in input sentences. Experimental results show that the pre-trained language models fine-tuned with the two proposed tasks can lead to improvement on five commonsense reasoning benchmark datasets.

Strengths:

+ The idea of teaching language models through self-supervised learning tasks is neat.

+ The performance of the proposed methods on few training examples looks great.

+ The results section is well structured. There are ablation studies on the training objectives of each proposed task as well as a comparison of generated sentences.


Concerns:

- The key concern about the paper is the lack of rigorous experimentation to study the effectiveness of the two self-supervised learning tasks. First, the methods are only compared with T5-base related methods on the four commonsense classification tasks. The leaderboard of commonsense QA shows that more than 20 systems report an accuracy higher than 63.32, which is the best configuration of the proposed method. Second, the proposed tasks are applied only to T5. I am wondering if it is effective on the other pre-trained language models. Third, the performance improvement on the classification tasks appears marginal. Statistical tests are desirable to show if such improvements are significant.

- Noun and verb phrases extracted from sentences are not always concepts. Masking-out certain words in the input is similar to the idea of removing non-content words from input. A deeper analysis of the proposed method would have been nice to understand which part is effective in the new task, keeping content words or not using mask-out, what if only concepts from a knowledge base is kept instead of content words?

-  The CALM model proposed in this work performs worse than the SOTA models in three out of four metrics on CommonGen, despite it uses less model parameters. What if the proposed tasks are applied on T5-Large?

Minor comments:

* I am not convinced that the generated corrupted sentences are in general grammatically correct, as stated in Sec. 2.1,

* I do not see a strong connection between completing COR and compositional reasoning, as stated in Sec. 2.1.

* The way of getting distractor sentences appears ad-hoc, may need further justification.

* Y in equation (5) and (6) needs explanation.

---

> ### Author Response · Authors · 2020-11-15
> **Response to Reviewer 4 (1/3)**
>
> Thanks for your insightful review and valuable feedback!
>
> >Weakness 1:
> >
> >The key concern about the paper is the lack of rigorous experimentation to study the effectiveness of the two self-supervised learning tasks. First, the methods are only compared with T5-base related methods on the four commonsense classification tasks. The leaderboard of commonsense QA shows that more than 20 systems report an accuracy higher than 63.32, which is the best configuration of the proposed method. Second, the proposed tasks are applied only to T5. I am wondering if it is effective on the other pre-trained language models. Third, the performance improvement on the classification tasks appears marginal. Statistical tests are desirable to show if such improvements are significant.
>
> It is true that the performance of base-size CALM and T5 is far from SOTA. However, it is notable that the SOTA models have 11B parameters, which is around 50 times more than our base size models.
>
> To test whether our proposed method can generalize to larger models, we continually pre-trained CALM-large with the T5-large model as the backbone after the submission deadline. The result comparison between CALM-large and T5-large is shown as follows (also see Table 2 in the revised draft).
>
> - Results of discriminative tasks
> |     method                  | #parameters      |    CSQA    |    OBQA   |   PIQA   |    aNLI    |
> | ---------------------------   |   :-------:   |   :--------:   | :-------: | :-------: | :-------: |
> |     T5-large                 |      774M                | 69.81  |  61.40  |  72.19   | 75.54 |
> |     CALM-large	       |      774M                | 71.31  |  66.00  |  75.11   | 77.12 |
> |     BERT-large            |      345M                | 57.06  |  60.04  |  67.08   | 66.75 |
> |     RoBERTa-large     |      345M                | 71.81  |  63.90  |  76.90   | 82.35 |
> |     SOTA                      |      11B                   | 79.1    |  87.2    |  90.13   | 89.70 |
>
> - Results on CommonGEN
> |method                                 | # parameters    |  BLEU4|  METEOR | CIDEr | SPICE |
> | ---------------------------   |   :-------:   |   :--------:   | :-------: | :-------: | :-------: |
> |BART                                      |  406M                 |  26.30  |  30.90       | 13.92  | 30.60 |
> |T5-large                                 |   774M                |  28.60  |  30.10       | 14.96  | 31.60 |
> |CALM-large	                       |   774M                |  29.50  |  31.90       | 15.61  | **33.20** |
> |KG-BART (SOTA, with KG)   |  406M                 |  **30.90**  |  **32.40**       | **16.83**  | 32.70 |
>
> We can see that CALM-large consistently improves upon T5-large on all tested datasets (statistical significantly with p-value < 0.01). The improvement is consistent with the improvement of CALM-base upon T5-base. This shows that our method can hopefully generalize well to even larger models like T5-3B and T5-11B.
>
> In addition, we have conducted statistical tests and find **the improvement of CALM (base size) upon T5-base to be significant with p-value < 0.01 on all classification datasets**. Also, we believe that the performance comparison of base size pre-trained models is meaningful because for SOTA models that contains 11B parameters: (1) the computational cost for continually pre-training them with our method is very computationally expensive (it will take around one month on 8xV100s), and (2) they are unlikely to be used in real-world applications because T5-3B and T5-11B model can’t be fitted in the GPU memory for inference even batch size set as 1 and the inference latency would be unsatisfactory. This is also suggested by the fact that the most popular (in terms of download times at huggingface’s model hub) pre-trained models are mostly base size models.
>
> We have also tested the effectiveness of CALM on other tasks and datasets to make the empirical results more convincing (See General Response NewTable 4). We can see CALM consistently outperforms the T5-base model across a set of knowledge-related tasks.

---

> > ### Author Response · Authors · 2020-11-15
> > **Response to Reviewer 4 (2/3)**
> >
> > As for other pre-trained models, we have continually pre-trained BART-base models with our objectives and tested its performance on different datasets. The results are shown as follows. We can see that our approach consistently and significantly (with p-value < 0.01) improves BART-base on all datasets. This result shows that our method is versatile to different pre-trained models.
> >
> > - Results with BART-base as backbone model (139M parameters)
> > |method            | CSQA | OBQA  |   PIQA  | aNLI   |  CommonGEN                 |
> > | ----------- | :-------: | :--------: | :-------: | :-------: | :-------: |
> > |BART-base      | 56.31| 58.30 |  67.53|59.85 |  25.10/29.50/13.16/30.20  |
> > |CALM-bart	 | **58.22**| **59.10** |  **69.40**|**61.28** |  **26.40**/**29.90**/**13.71**/**31.10**  |
> >
> > >Weakness 2:
> > >
> > >Noun and verb phrases extracted from sentences are not always concepts. Masking-out certain words in the input is similar to the idea of removing non-content words from input. A deeper analysis of the proposed method would have been nice to understand which part is effective in the new task, keeping content words or not using mask-out, what if only concepts from a knowledge base are kept instead of content words?
> >
> > That’s a good point. We have tested the variant for the C2S objective where only concepts from ConceptNet are kept. The performance on CSQA, and PIQA is lower than the version used in our paper, as shown in the Table below:
> >
> > |method                                                 |CSQA |   PIQA  |
> > | ----------- | :-------: | :--------: |
> > |CALM-C2S (ours: nouns + verbs)                 | **62.24**| **68.75** |
> > |CALM-C2S (concepts in ConceptNet)  |61.88 | 68.29  |
> >
> > We can see that our method outperforms this variant. We find that when only keeping concepts in ConceptNet, the input concepts are fewer and make the input information not enough for recovering the original sentence. Specifically, for more than 40% of the sentences, the concept-set extracted by ConceptNet is smaller than that of all content words by at least 2 concepts.
> >
> > Also, we have conducted an ablation study about using either verbs or nouns as concepts. The results are shown as follows:
> >
> > |method            | CSQA | OBQA  |   PIQA  | aNLI   |  CommonGEN                   |
> > | ----------- | :-------: | :--------: | :-------: | :-------: | :-------: |
> > |CALM              | **63.32**| **60.90** |  **71.01**|**63.20** |  **26.40**/**31.40**/**13.88**/**33.00**  |
> > |CALM-noun	 | 62.45| 59.40 |  69.05|61.55 |  25.70/31.20/13.17/32.60  |
> > |CALM-verb	 | 62.51| 59.10 |  69.24|61.40 |  25.60/31.20/13.24/32.60  |
> >
> > We can see that using either nouns-only or verbs-only as concepts for our approach leads to a substantial performance drop. This supports our choice of using both nouns and verbs as concepts.
> > In addition, our objectives are not merely about “masking” and denoising, but they compose sentences with concepts.
> >
> > >Weakness 3:
> > >
> > >The CALM model proposed in this work performs worse than the SOTA models in three out of four metrics on CommonGen, despite it uses less model parameters. What if the proposed tasks are applied on T5-Large?
> >
> > We have conducted experiments on T5-large and the results are shown in response 1. With T5-large, CALM-large’s performance is comparable with the SOTA model (KG-BART).  Also, KG-BART exploits an external knowledge base for fine-tuning and inference. In contrast, our method does rely on any external KB and make it easier to use in practice. In addition, CALM-large outperforms SOTA on the SPICE metric, which is the most important metric on CommonGEN because it correlates the best with human evaluation, according to section 5.1 in the original paper of CommonGEN (Lin et al. 2020).
> >
> > ==============================\
> > Reference:\
> > CommonGen: A Constrained Text Generation Challenge for Generative Commonsense Reasoning  Bill Yuchen Lin, Wangchunshu Zhou, Ming Shen, Pei Zhou, Chandra Bhagavatula, Yejin Choi and Xiang Ren  In Findings of EMNLP 2020

---

> > > ### Author Response · Authors · 2020-11-15
> > > **Response to Reviewer 4 (3/3)**
> > >
> > > >Concern 1:
> > > >
> > > >I am not convinced that the generated corrupted sentences are in general grammatically correct, as stated in Sec. 2.1,
> > >
> > > A large portion of the generated corrupted sentences are grammatically correct because nouns are only shuffled with nouns and verbs are shuffled with verbs. It is true that some of them are not grammatically correct because of issues in verb forms. However, we do not have to ensure the sentence to be ALWAYS grammatically correct because our objective aims to denoise even from corrupted sentences and we believe that the majority of noise in the corrupted sentences is the order and relation between concepts.
> > >
> > > >Concern 2:
> > > >
> > > >I do not see a strong connection between completing COR and compositional reasoning, as stated in Sec. 2.1.
> > >
> > > Thanks for pointing it out. The COR task is more related to relational reasoning because it requires the model to correct the order between different concepts, which involves understanding and reasoning the relation between different concepts. It is the C2S task that is related to compositional reasoning because it requires the model to compose sentences with different compositions of concepts, according to (Lin et al. 2020). In the revised draft, we have corrected the claim.
> > >
> > > >Concern 3:
> > > >
> > > >The way of getting distractor sentences appears ad-hoc, may need further justification.
> > >
> > > Generating distractors for contrastive learning in a GAN-style is related to previous work for language model pre-training like ELECTRA, which uses a masked language model to generate distractor tokens. This choice makes the distractor hard to distinguish and makes the training more informative. In addition, we chose to use the sentences generated by the model itself as distractors because the model exploits the knowledge that is already packed in its parameters to generate them. Therefore, training the model to distinguish them will force the model to acquire new commonsense knowledge that it lacks. In this way, the model is trained to iteratively improve upon itself in a self-play fashion. We think it may be a good choice and further investigation can be left for future work.
> > >
> > > >Concern 4:
> > > >
> > > >Y in equation (5) and (6) needs explanation
> > >
> > > We’ve added explanations in the revised version. Thanks for pointing this out!

---

> > > > ### Author Response · Authors · 2020-11-20
> > > > **Look Forward to Hearing From You**
> > > >
> > > > Dear Reviewer 4,
> > > >
> > > > We appreciate your valuable feedback and comments.
> > > > We could improve our paper in a good way with your great feedback!!
> > > >
> > > > We want to send you a friendly reminder that the first phase of the response period is completed, and want to hear back from you about our response.
> > > >
> > > > If you could let us know any other concerns that we could not address in the response, we'd be happy to talk about it!
> > > > Thanks again for your time, and your response is eagerly awaited.
> > > >
> > > > Thanks.

---

> > > > > ### Author Response · Authors · 2020-11-23
> > > > > **Reminder for the discussion**
> > > > >
> > > > > Dear Reviewer 4,
> > > > >
> > > > > We want to send you a friendly reminder for the discussion.
> > > > > Here are the things that we have added and resolved by your valuable feedback!
> > > > >
> > > > > - We include results of **CALM-large** which is based on T5-large. The results show improvement.
> > > > > - We include results of **CALM(BART)**, which is by applying our method on BART. The results show the effectiveness of our method on other pertained language models.
> > > > > - To check the **effectiveness of noun and verb phrases as concepts**, we conducted an additional experiment with only concepts from ConceptNet. The results show that noun and verb phrases as concepts is more effective than concepts from ConceptNet.
> > > > >
> > > > > We thank you again for your valuable comments, and we would appreciate it if you could review our improvement and give us another round of feedback.
> > > > >
> > > > > Thanks.

---

### Author Response · Authors · 2020-11-15
**Response to All Reviewers / Revision Details (1/2)**

Hi all. We’ve revised and updated the draft according to your valuable reviews. The updates are summarized as follows:
- Included results of CALM-large which is based on T5-large. (See Table 2 in the revised draft or NewTable-1 in this comment)
- Included results of CALM (BART-base), which is obtained by applying our method on BART-base. The experiments with BART-large are currently running. (See Table 9 in the revised draft or NewTable-2 in this comment)
- Included results of ablation study analyzing the choice of using either nouns or verbs as concepts. (See Table 10 in the revised draft or NewTable-3 in this comment)
- Included experiments on LAMA probes  (Petroni et al.,2019)  and KILT  (Petroni et al.,2020), two sets of benchmark datasets for knowledge-intensive tasks (See Table 6 in the revised draft or NewTable-4 in this comment).
- Included results of several popular base size pre-trained language models in Table 1 of the revised draft and conducted a statistical significance test. Results show that CALM significantly outperforms T5-base, BERT-base, ERNIE, and KnowBERT with a p-value < 0.01.
- We conducted a human evaluation of CommonGEN predictions.
- We’ve polished the writing and figures in the paper.
- **Nov 24 paper updates** The paper is updated with all the additional experiments we have conducted during the rebuttal phase.

First of all, after the submission deadline, we have continually pre-trained CALM-large with T5-large as the backbone model and now the results are available (see Fig 2 in the revised paper):

- Results of discriminative tasks
|     method                  | #parameters      |    CSQA    |    OBQA   |   PIQA   |    aNLI    |
| ---------------------------   |   :-------:   |   :--------:   | :-------: | :-------: | :-------: |
|     T5-large                 |      774M                | 69.81  |  61.40  |  72.19   | 75.54 |
|     CALM-large	       |      774M                | 71.31  |  66.00  |  75.11   | 77.12 |
|     BERT-large            |      345M                | 57.06  |  60.04  |  67.08   | 66.75 |
|     RoBERTa-large     |      345M                | 71.81  |  63.90  |  76.90   | 82.35 |
|     SOTA                      |      11B                   | 79.1    |  87.2    |  90.13   | 89.70 |

- Results on CommonGEN
|method                                 | # parameters    |  BLEU4|  METEOR | CIDEr | SPICE |
| ---------------------------   |   :-------:   |   :--------:   | :-------: | :-------: | :-------: |
|BART                                      |  406M                 |  26.30  |  30.90       | 13.92  | 30.60 |
|T5-large                                 |   774M                |  28.60  |  30.10       | 14.96  | 31.60 |
|CALM-large	                       |   774M                |  29.50  |  31.90       | 15.61  | **33.20** |
|KG-BART (SOTA, with KG)   |  406M                 |  **30.90**  |  **32.40**       | **16.83**  | 32.70 |
- NewTable 1

We find that CALM-large **significantly improves upon T5-large with p-value < 0.01 on all datasets**. This confirms that our approach is effective for larger models, which may help resolve the concern of R3 and R4. Also, CALM-large significantly outperforms BERT-large while performs slightly worse than RoBERTa, which is optimized with more gradient updates and cannot be used for NLG tasks. On CommonGen, CALM-large yields state-of-the-art performance on the SPICE metric, which correlates best with human evaluation according to the CommonGEN paper.

- Results with BART-base as backbone model (139M parameters)
|method            | CSQA | OBQA  |   PIQA  | aNLI   |  CommonGEN                 |
| ----------- | :-------: | :--------: | :-------: | :-------: | :-------: |
|BART-base      | 56.31| 58.30 |  67.53|59.85 |  25.10/29.50/13.16/30.20  |
|CALM-bart	 | **58.22**| **59.10** |  **69.40**|**61.28** |  **26.40**/**29.90**/**13.71**/**31.10**  |
- NewTable 2

We can see that our approach consistently and significantly (with **p-value < 0.01**) improves BART-base on all datasets. This result shows that our method is versatile to different pre-trained models.

---

> ### Author Response · Authors · 2020-11-15
> **Response to All Reviewers / Revision Details (2/2)**
>
> - Results with Noun / Verb as a concept
>
> |method     | CSQA  | OBQA  |   PIQA  | aNLI   |  CommonGEN      |
> | -----------   | :-------: | :--------: | :-------: | :-------: | :-------: |
> | CALM              | **63.32** | **60.90** |  **71.01**|**63.20** |  **26.40**/**31.40**/**13.88**/**33.00** |
> | CALM-noun	 |  62.45 | 59.40 |  69.05|61.55 |  25.70/31.20/13.17/32.60  |
> | CALM-verb	 | 62.51| 59.10 |  69.24|61.40 |  25.60/31.20/13.24/32.60  |
>
> - NewTable 3
>
> We can see that using either nouns-only or verbs-only as concepts for our approach leads to a substantial performance drop. This supports our choice of using both nouns and verbs as concepts.
>
>
> - LAMA Probe
> | method    | MRR  | Precision@50| Precision@10| Precision@1|
> | ----------- | :-------: | :--------: | :-------: | :-------: |
> |T5-base   |11.53  |  38.52             |     21.60         |         5.93      |
> |CALM	    | **12.09** |  **39.69**             |     **22.53**         |     **6.46** |
>
> - KILT tasks (acc)
> |method | FEVER (fact verification )| AY2 (entity linking) |
> | ----------- | :-------: | :--------: |
> |T5-base| 76.65                              | 74.97                      |
> |CALM   | **77.44**                               | **77.24**        |
>
> - NewTable 4
>
> We can see CALM consistently outperforms the T5-base model across a set of knowledge-related tasks.
>
> For human evaluation, We asked three annotators to choose the most reasonable sentence between T5-base and CALM-base predictions.
> Here's the detail:
> - Number of test sentences: 50
> - Binary selection by majority voting
> - Kappa score (inter-annotator agreement) : 0.73
> - CALM-base : 0.6 / T5-base : 0.4
>
> ==============================
>
> Reference: \
> Petroni et al.,2019 Language Models as Knowledge Bases?, EMNLP 2019\
> Petroni et al.,2020 Kilt: a benchmark for knowledge intensive language tasks, arxiv

---

### Decision · Program_Chairs · 2021-01-07
**Final Decision**

**Decision:**

Accept (Poster)

**Comment:**

This paper presents two self-supervised learning objectives that can be used as intermediate pre-training tasks to refine the T5 sequence-to-sequence model between pre-training and task fine-tuning. It shows that, at small to moderate model sizes, adding this step significantly improves performance on commonsense-oriented target tasks.

Pros:
- This appears to be a fairly straightforward improvement in self-supervised learning in NLP, with fairly extensive experiments.

Cons:

- This model isn't trained at the same extremely large scales (10B+-words) as state-of-the-art models, and it performs significantly below the state of the art. It's not clear that the released model represents a useful model for any application as-is, and while it's likely, it's not proven that the ideas in this paper would still be useful at larger scales.
- Given that, it seems like the most likely audience for this work is other developers of pretrained models in NLP, which makes the fit to a general ML conference less clear.
- The framing around 'concepts' and, more importantly, the model name 'concept-aware LM', gives the unwarranted impression that the new model handles 'concepts' in a way that T5 doesn't. It is not reasonable to use the word 'concept' to refer to specific parts of speech in your title (even if you later explain that), and whether your model handles concepts in a categorically different way from T5 would take a substantial analysis to show, which doesn't seem to be present. I don't think this paper is up to ICLR's standards with the current name, and urge the authors to change it.